# Repeated Thermal Shock Behavior of ZrB_2_–SiC–Graphite Composite under Pre-Stress in Air and Ar Atmospheres

**DOI:** 10.3390/ma13020370

**Published:** 2020-01-13

**Authors:** Xing Yue, Xianghe Peng, Zhen Wei, Xiaosheng Chen, Xiang Chen, Tao Fu

**Affiliations:** 1College of Aerospace Engineering, Chongqing University, Chongqing 400044, China; xingyue@cqu.edu.cn (X.Y.); futao@cqu.edu.cn (T.F.); 2State Key Laboratory of Coal Mining Disaster Dynamics and Control, Chongqing University, Chongqing 400044, China; 3Department of Mechanical Engineering, Tokyo Institute of Technology, Tokyo 152-8552, Japan; xschen@cqu.edu.cn; 4Advanced Manufacturing Engineering, Chongqing University of Posts and Telecommunications, Chongqing 400065, China; chenxiang@cqupt.edu.cn

**Keywords:** ultra-high temperature ceramics, repeated thermal shock, Ar atmosphere, pre-stress, mechanical properties

## Abstract

The thermo–chemo–mechanical coupling on the thermal shock resistance of 20 vol%-ZrB_2_–15 vol%-SiC–graphite composite is investigated with the use of a self-developed material testing system. In each test, a specimen under prescribed constant tensile pre-stress (*σ*_0_ = 0, 10, 20 and 30 MPa) was subjected to 60 cycles of thermal shock. In each cycle, the specimen was heated from room temperature to 2000 °C within 5 s in an air atmosphere or an Ar atmosphere. The residual flexural strength of each specimen was tested, and the fracture morphology was characterized by using scanning electron microscopy (SEM). There were three different regions in the fracture surface of a specimen tested in the air, while no such difference could be observed in the fracture surfaces of the specimens that were tested in Ar. The residual flexural strength of the composite that was tested in Ar generally decreases with the increase of *σ*_0_. However, in the range of 0 ≤ *σ*_0_ ≤ 10 MPa, the residual flexural strength of the composite that was tested in the air ascended with the increase of *σ*_0_ due to the healing effect of oxidation, but it descended thereafter with a further increase of *σ*_0_, as the effect pre-stress that became prominent.

## 1. Introduction

Zirconium diboride (ZrB_2_) is an important member of ultrahigh-temperature ceramics (UHTCs) [1] that has been extensively used as wing leading edges, nose-caps and propulsion system components in space vehicles due to its excellent thermal, mechanical, and chemical properties, such as its high melting point (>3000 °C), high electrical and thermal conductivity, high corrosion resistance, and chemical stability [2,3,4,5,6,7,8,9,10]. Among these properties, thermal shock resistance (TSR) is of vital importance, as most nose-cap, leading edge and aero-engine applications re inevitably subjected to thermal shock, such as the sudden temperature change by high-speed friction with air. Additionally, an appropriate strength, thermal expansion coefficient, density and machinability are equally important for the application of UHTCs [11,12,13].

At present, most of the methods that are used for testing the TSR of UHTCs are based on quenching thermal shock and the temperature difference at which the strength of a material after quenching decreases to 70% of its initial strength is defined as the critical temperature difference (Δ*T*_c_) of quenching thermal shock. Because of its low cost, easy operation, simplicity and maturity, this method has been widely used to investigate the TSR of UHTCs [14,15,16,17,18,19]. Jin investigated the effects of porosity and pore size on the mechanical properties and thermal shock fracture resistance of a sintered ZrB_2_–SiC composite with a single cycle of room-temperature water quenching and found that porous ZrB_2_–SiC samples possessed higher thermal shock residual strength than dense ZrB_2_–SiC samples [16,20]. Hu performed the water quenching thermal shock of a ZrB_2_–SiC–graphite composite heated in a vacuum and an air atmosphere and showed that the residual strength of the samples heated in air had a lift when the temperature change (Δ*T*) was higher than 1200 °C, while the residual strength of the samples heated in a vacuum decreased with the increase of Δ*T* [21,22,23]. Chen [17,24] investigated the effect of the gradient of surface heat transfer coefficient on the strength of a ZrB_2_–SiC–graphite composite that was subjected to thermal shock that was induced by quenching in different mediums—aqueous silicone oil, water, and boiling water. Zhang et al. systematically investigated the thermal shock behavior of diboride-based UHTCs with additions such as carbon black, an SiC whisker, and disilicide (TaSi_2_) [25,26,27,28]. Furthermore, the TSR of composites with additions of graphite, SiC fibers, SiC nanoparticles, carbon nanotubes, and carbon fiber were also extensively investigated [7,29,30,31,32,33,34,35].

It is hard for people to perform repeated thermal shock with water quenching experiments at a large Δ*T* (>2000 °C) by heating specimens with a Murphy furnace. On the one hand, heating a specimen to a very high temperature usually takes a long time, and most furnaces can hardly work at such a high temperature. On the other hand, the cooling rate generated by quenching is usually extremely large and even beyond the range of actual applications. There is a problem when using thermal shock induced by quenching to simulate that induced by fast-heating, because their thermal–mechanical processes, the surrounding environments, the changes in material property, and the modes of deformation and failure might be completely different. Fortunately, there is another means to more realistically achieve the actual thermal shock condition. In fact, for UHTCs with a satisfactory conductivity, such as ZrB_2_, it is more convenient to use the resistance heating method to simulate thermal shock that is related to fast-heating, where the interactions between components and the varying atmospheres around flight and re-entry conditions can be more realistically considered [36,37]. For example, Meng et al. systematically performed the repeated thermal shock of ZrB_2_–SiC and ZrB_2_–SiC–ZrC composites with resistance heating, and they showed that the residual strength did not decline monotonically with the increase of thermal shock cycles that reuslted from the formation of the pores in the oxide layer during thermal shocks [38,39,40,41]. In their work, the effect of pre-stress and environmental atmospheres, which is usually encountered in the actual application of UHTCs, was not involved. Jin et al. investigated the effect of heating rate on the residual strength of a ZrB_2_–SiC–graphite (ZSG) composite in atmospheres of low oxygen partial pressure (200 Pa) and in pure oxygen [42,43], and they revealed that the TSR is sensitive to the heating rate; the residual strength decreased with the increase of Δ*T* as 1800 °C < Δ*T* < 2200 °C. Meanwhile, the residual strength decreased with the increase of heating rate for the same Δ*T*. They also showed that the size effect on the TSR with resistance heating was different from that with quenching; the specimens of larger size exhibited a lower residual strength. However, the effect of pre-stress on the residual strength of materials that are subjected to thermal shock, as well as the effect of oxidation, is still unclear. Though some progress has been made in the thermal fatigue life of ZSG composites under pre-stress [44,45], the effect and the mechanism of pre-stress on the accumulated damage of the composites that are subjected to thermal shock in different atmospheres need more detailed investigation, considering that residual flexural strength is usually affected by complex effects of coupled thermal shock and oxidation and keeping in mind that pre-stress exists in most actual engineering structures.

In this article, we intend to separately investigate the effects of pre-stress and oxidation on the mechanical properties of a 20 vol%-ZrB_2_–15 vol%-SiC–graphite composite that is subjected to repeated thermal shock. In Section 2, the experimental procedure is briefly introduced, and the residual flexural strengths of the specimens are tested. In Section 3, the microstructures of the tested specimens are analyzed with a scanning electron microscope (SEM), and the accumulated damage (microcracks) that are caused by repeated thermal shock under various pre-stresses in air and in Ar environments is analyzed. Some conclusions are drawn and given in Section 4.

## 2. Experimental Procedure

The schematic diagram of the self-developed, coupled thermal–mechanical material testing system used for our tests is shown in Figure 1. Its physical diagram and details were also introduced elsewhere [44,45,46]. A lambda sensor (7OVX, City Technology, Ltd., Hampshire, UK) was equipped in the chamber to measure the oxygen content during the tests.

### 2.1. Specimen Preparation

The 20 vol%-ZrB_2_–15 vol%-SiC–graphite composite (ZSG) that was used in this experiment was provided by Beijing Zhongxin Technology Co., in which the powders of ZrB_2_ (Beijing Zhongxin Technology Co., Ltd., Beijing, China), SiC (Weifang Kaihua Micropowder Co., Ltd., Weifang, China) and graphite flake (Qingdao Tiansheng Graphite Co., Ltd., Qingdao, China) had purities of over 99.0%, and the mean diameters of ZrB_2_ and SiC were 2 and 0.5 µm, respectively. The mean diameter and the thickness of the graphite flake were 15 and 1.5 µm, respectively. The volume fractions of the three kinds of powders in the composite were 0.65, 0.20 and 0.15, respectively. The powder mixtures were ball-milled at a speed of 220 rpm for 12 h, with ZrO_2_ balls and ethanol as the grinding media. After mixing, the slurry was dried in a rotary evaporator and screened. Then, the mixtures were at 1900 °C in a furnace (FVPHP-R-5, Fuji Dempa Kogyo Co., Ltd., Tokyo, Japan) in a vacuum for 60 min under a uniaxial compressive stress of 30 MPa while using a boron nitride coated graphite die. The relative density of the bulk material was 96.3%. The specimens were strips of 2 × 3 × 36 mm (width × height × length), with their length parallel to the direction of the compressive stress applied during sintering.

### 2.2. Repeated Thermal Shock Tests

During the repeated thermal shock experiments, the specimens were subjected to constant tensile stresses of *σ*_0_ = 0, 10, 20 and 30 MPa, which were provided by standard weights. Two different gaseous atmospheres were used for the tests: air and Ar atmospheres. For the latter, the chamber was sealed with a cover, and the gas inside was pumped out with a vacuum pump (Trivac D8C, Oerlikon Leybold Vacuum Co. Ltd., Tianjing, China) until the vacuity in the vacuum chamber was lower than 10 Pa. Then, the Ar gas (with a purity of over 99.999%) was fed until it fully filled the chamber. Such extraction and feeding of Ar gas was repeated three times before thermal shock testing started. The oxygen content in the sealed chamber measured with a lambda sensor (7OVX, City Technology, Ltd., Hampshire, UK) is shown in Figure 2, which indicates that the final oxygen content in the vacuum chamber was less than 1500 ppm. In each test, a specimen was heated from room temperature (25 °C) to 2000 °C by applying a electrical current of 350 A, which was then automatically switched off when the temperature at the center of the specimen surface reached 2000 °C. The variations of real-time temperature and electric current against the testing time between two cycles of thermal shock is displayed elsewhere [44]. In each cycle, the time for applying the current was about 5 s, and then the specimen was cooled down naturally for 25 s, which was found to be sufficient for the temperature in the specimen to fall back to room temperature (25 °C) from 2000 °C. The average heating rate was about 500 °C/s from 1000 to 2000 °C, and the average cooling rate was about 300 °C/s from 2000 to 1000 °C. In order to show the accumulated effects of the adopted experimental conditions and to prevent the specimen from failure during the repeated thermal shock test, in each test, the number of thermal shock cycles was limited to 60. The experimental conditions are listed in Table 1, and for each condition, at least three specimens were tested.

### 2.3. Flexural Tests

The residual flexural strength of each specimen after 60 cycles of thermal shock was tested by an instrument (EZ-LX, Shimadzu, Inc. Japan) with a crosshead speed of 0.1 mm/min. The following equation was used for calculating the residual flexural strength:(1)$$\sigma_{f}=\frac{3FL}{2bh^{2}}$$
in which *L*, *b*, *h* and *F* are the fixture span, width, height, and the failure load of specimen, respectively. Considering the capacity of testing equipment, *L =* 25 mm, *b* = 3 mm and *h* = 2 mm were chosen. The results of the tests are shown and discussed in Section 3.2.

## 3. Results and Discussion

### 3.1. Surface Analysis

The surfaces of the specimens that were tested with different pre-stresses in the air or Ar atmospheres are shown in Figure 3. It can be seen that there was a significant difference between the surfaces of the specimens that were tested in two different kinds of gaseous atmospheres. Due to oxidation, the central parts of the surfaces of the specimens that were tested in air became white, while the parts near the two ends looked dark. On the contrary, the central parts of the surfaces of the specimens that were tested in the Ar atmosphere were gray, and both sides near the central parts looked white. The constituent on the surface of specimens that were tested in air was mainly borosilicate glass and ZrO_2_ [47], while that of specimens that were tested in the Ar atmosphere was mainly ZrO_2_ (as shown in Figure 4) which can be attributed to the very low oxygen content (<1500 ppm); this information was obtained by using energy dispersive spectroscopy (EDS) analyses. Additionally, the occurrence of a minor amount of hafnium could be ascribed to the imported contamination during material preparation. For the specimens that were tested under *σ*_0_ = 20 MPa in the air and in Ar atmospheres, the surface images at 2000 °C that were taken by the camera are shown in Figure 5, where bubbles can be seen on the specimen that was tested in air due to the escape of the gaseous oxidation products beneath the surface glassy layer, such as CO, SiO and B_2_O_3_. However, on the surface of the specimen that was tested in the Ar atmosphere, there were only a few, almost invisible tiny bubbles. The surfaces of the specimens that were tested in air were rough due to the appearance of bubbles, while the surface of the specimen that was tested in the Ar atmosphere looked much smoother. Figure 6a shows the SEM photographs of the surface of the specimen that was subjected to 60 cycles of thermal shock in air under *σ*_0_ = 20 MPa, and it can be seen that the growth of the oxide products (ZrO_2_) exhibited a ridge-valley-like feature, where liquid oxide products (SiO_2_ and a small amount of dissolved B_2_O_3_) flowed along the valleys, as shown in Figure 6a and its inset. The ripple on the surface provides distinct evidence of an oxidation reaction and the existence of borosilicate glass layer. For the specimen that was tested in the Ar atmosphere, as shown in Figure 6b, there was no such outermost borosilicate glass layer. The outermost layer was mainly composed of ZrO_2_ particles.

Figure 7 and Figure 8 show the SEM photographs of the fracture surfaces of the specimens that were tested in different atmospheres after 60 cycles of thermal shock under *σ*_0_ = 0, 10, 20 and 30 MPa, followed by bending fracture tests. Figure 7 shows the fracture surfaces of the specimens that were tested in air; in each of these, there are three parts: the glassy layer, the oxide layer and the substrate. The boundaries between these parts were distinct, as indicated by the dashed lines. However, in the specimens that were tested in the Ar atmosphere, as shown in Figure 8, there was no distinct glassy layer or oxide layer; there was mainly a substrate that was covered by a thin surface layer, and the boundary between them could be found. This indicates that for the specimens that were tested in the Ar atmosphere, a chemical reaction played an insignificant role, i.e., the oxidation and tis corresponding effect on the material property could be ignored.

It can be seen in Figure 7 that the thicknesses of the outermost glassy layers of the specimens for ZSG0, ZSG10, ZSG20, and ZSG30 were 4.98, 4.04, 5.64, and 4.80 µm, respectively, of which the average thickness was approximately 4.87 µm. These glassy layers were mainly composed of solidified SiO_2_ with a small amount of dissolved B_2_O_3_, both which are the oxides of SiC and ZrB_2_. Each cycle of thermal shock could be divided into two processes: When heating up, the gaseous oxide of B_2_O_3_, CO and SiO would have been transmitted to the specimen surface, where a large amount of liquid SiO_2_ would have been formed. Because the heating time would have been very short, most gaseous oxides would have diffused into air, and a little B_2_O_3_ would have dissolved in the liquid SiO_2_ and remained on the specimen surface. During cooling, this mixture would have solidified to form an amorphous glassy layer on the specimen surface that would have reduced the diffusion of oxygen into the specimen and would have provided effective protection against oxidation until it reached a temperature of 1800 °C. However, with the further increase of thermal shock cycles, the oxidation time would have increased, and the amorphous glassy layer would have tended to completely evaporate, resulting in a porous ZrO_2_ surface layer [47,48].

The oxidation product, ZrO_2_, would have formed the skeleton in the oxide layer. The grains in this layer were different from those in the substrate. The grains in the substrate were of similar equiaxial shape, while the grains in the oxide layer were columnar. It is known that ZrO_2_ can be dissolved in the liquid mixture of liquid borosilicate, and, with the evaporation of gaseous oxide, ZrO_2_ precipitates and forms columnar grains. On the other hand, the interstices between columnar grains also serve as the pathways for oxygen to diffuse inwards and for gaseous oxide to diffuse outwards, which could promote the oxidation in the composite. As a consequence, a solidified mixture would have dispersed between the columnar ZrO_2_ grain boundaries. The thicknesses of the oxide layers corresponding to *σ*_0_ = 0, 10, 20 and 30 MPa were 23.61, 24.82, 23.98 and 23.42 µm, respectively, as shown in Figure 7. This shows that, in our previous thermal shock experiments with UHTCs [44,46], with sufficient oxidation time, the grains in this layer could have had a columnar shape and the thickness of this layer could have been large; this would have been less dependent on the applied pre-stress under the adopted experimental conditions.

The thicknesses of the mixed layers, including the oxide and outermost glassy layers of ZSG0, ZSG10, ZSG20, and ZSG30 were 28.59, 28.86, 29.62 and 28.22 µm, respectively, with an average thickness of 28.82 µm, which was much thicker than that of the surface layers of the specimens that were tested in the Ar atmosphere. Instead of the outmost glassy layer, the surface layer of the specimen tested in the Ar atmosphere was slightly oxidized and was composed of a minor amount of ZrO_2_ particles. Compared with the composite that was tested in air, there was no borosilicate and dissolved SiO_2_ in the oxide layer, as shown in Figure 4, due to the very limited content of oxygen. In Figure 8, the grains in the surface layer can be seen to be almost equiaxial, similar to the grains in the substrate and the intergranular pores that can be observed from each fracture surface. The pull-out of graphite flakes can also be observed, as marked by the red ellipses in Figure 8, which indicates the weak bond between the graphite flake and substrate. On the contrary, few graphite flakes could be found in the mixed layers of the specimens that were tested in air, as shown in Figure 7. These differences could be ascribed to the different atmospheres and the severe oxidation of graphite flakes that was experienced in air at a temperature over 500 °C [8]. The pores vanished due to the growth of ZrO_2_ grains and the solidified borosilicate being dispersed between these grains.

The fractographs and corresponding backscattered electron images of the ZSG composite specimens that were tested in the air and Ar atmospheres under *σ*_0_ = 0, 10, 20 and 30 MPa are shown in Figure 9 and Figure 10, respectively. The sites that are indicated by black arrows are identified as the aggregate of SiO_2_ particles, which also refer to the pits on the fracture surface. As revealed in our previous investigation [45], under a small *σ*_0_, a fracture may take place due to the coalescence of a great number of microcracks that usually initiate from the area where SiC particles aggregate. While under a large *σ*_0_, the composite is inclined to form large microcracks, so a fracture might occur due to the continuous propagation of these large microcracks [45]. It can be seen in Figure 9 that the density of pits in the fractographs of ZSG20 and ZSG30 (Figure 9C,D) was larger than that of ZSG0 and ZSG10 (Figure 9A,B). Figure 10 shows more pits in the fractographs of the specimens that were tested in the Ar atmosphere compared to that of the specimens that were tested in air. This difference could be ascribed to the absence of a glassy layer, as, without this protective layer, oxygen could diffuse into the interior of the specimen more easily—although the oxygen content was very limited in the Ar atmosphere. In both testing environments, the density of the pits increased with the increase of *σ*_0_. A larger *σ*_0_, as well as a large number of pits, was more conducive to the formation of microcracks in the specimen. Moreover, the larger *σ*_0_ values were also beneficial for the propagation of the microcracks. We also observed that the pits were almost located at the boundaries of the specimens, which implies that microcracks were more easily initiated within the layer near the surface and then propagated to the inside of the specimen.

The typical microcracks in the fracture surfaces of the specimens that were tested in the air and Ar atmospheres under *σ*_0_ = 0, 10, 20 and 30 MPa are shown in Figure 11 and Figure 12, respectively. They show that the length of the microcracks, in general, increased with the increase of *σ*_0_. On the other hand, under the same *σ*_0_, the average length of the microcracks in the fracture surface of the specimen that was tested in air was larger than that of the specimen tested in the Ar atmosphere. Additionally, the maximum length of the microcracks in the fracture surface of the specimen that was tested in air was measured as 1048.9 µm (Figure 11D), while that of the specimen that was tested in the Ar atmosphere was about 561 µm (Figure 12C). It is known that the stress in the central parts of a specimen that is tested in air comes from three forces: thermal mismatch, oxidation and applied pre-stress, among which the first and the second parts are negative and positive, respectively. Therefore, the central part of the specimen might have been subjected to a relatively larger tensile stress, accounting for the relatively longer microcracks. On the other hand, for the specimen tested in the Ar atmosphere, the second part vanished due to a lack of oxidation, thus making the tensile stress relatively smaller and accounting for the relatively shorter microcracks. Additionally, the sites where graphite flakes were pulled out may have acted as the sources and propagation paths of microcracks, as shown in Figure 12A. Combined with the analysis for pits, there was a larger possibility for microcracks to initiate in the specimens that were tested in Ar, while there was a higher possibility for microcracks to propagate in the specimens that were tested in air due to a relatively larger tensile stress.

### 3.2. Residual Flexural Properties

The residual flexural strengths (*σ**_f_*) of ZSG0–Ar, ZSG10–Ar, ZSG20–Ar, and ZSG30–Ar were 401.5, 374.9, 393.1, and 369.8 MPa, respectively, all of which were significantly lower than the flexural strength of the specimen that was created with the fabricated composite, 489.4 MPa, as shown in Figure 13. In the tests in the Ar atmosphere, the oxidation of the specimen was insignificant and could be ignored. It could be seen that, in general, *σ**_f_* decreased with the increase of *σ*_0_. As can be seen, the microcracks in the specimens were much smaller without applying *σ*_0_, with the longest microcrack being 132.3 µm (Figure 12A). With the increase of *σ*_0_, the accumulated damage (in the form of the length of microcracks) increased, which, in turn, decreased the *σ**_f_* of the specimens. The graphite flakes only existed in this environment, and the pull-out sites could also have been the origin of the microcracks’ initiation, as mentioned in Section 3.1.

The *σ**_f_* values of ZSG0 and ZSG10 were 428.4 and 456.3 MPa, respectively, both of which were a little higher than those of the specimens that were tested in the Ar atmosphere due to oxidation, i.e., the healing effect of the low viscosity borosilicate glass layer that formed on the surface of the specimen [49], and the compressive stress in the oxide layer that resulted from volume expansion [38,39,50]. In this range of applied *σ*_0_, the healing effect played a dominant role, while the effect of *σ*_0_ was not obvious. Though ZSG10 had the largest *σ**_f_* among all the specimens that were tested, it was still lower than that of the specimen that did not suffer any thermal shock. The number and length of the microcracks that were generated under this condition were kept at a low level (Figure 11B), which proves that, at a low *σ*_0_, the healing effects created due to the oxidation reaction of ZrB_2_ and SiC could bring in effective protection and improve thermal shock resistance. With the increase of *σ*_0_, the effect of pre-stress became apparent, as *σ**_f_* decreased quickly. The *σ**_f_* of ZSG20 was very close to that of ZSG20–Ar, and the length of the microcracks on the fracture surface of the specimens that were tested in these two environmental conditions were also very similar, as shown in Figure 11C and Figure 12C. The *σ**_f_* of ZSG30 decreased to 362.4 MPa, which was even lower than that of ZSG30–Ar; this was consistent with the occurrence of the largest cracks (Figure 11D). Under this condition, oxidation could no longer yield a healing effect, accounting for the reduction of *σ**_f_*. The accumulated damage in each cycle of thermal shock should have been the result of coupled oxidation and thermal shock, which should have been larger than the accumulated damage that was caused by pure thermal shock. Under a larger *σ*_0_, the healing effects due to the compressive stress was overwhelmed by the large tensile stress, so new defects were consequently be generated, either in the oxide region or at the interface between the oxide layer and the substrate [51], as shown in the red ellipses in Figure 7c,d. With the chemical effect, it was be easier for the microcracks to propagate, grow, and coalescence under a larger *σ*_0_, as shown in Figure 11.

The variations of *σ**_f_* against *σ*_0_ of the specimens that were tested in the two atmospheres can be fitted with the following simple linear relationship:(2)y=ax+b
in which *x* is the applied pre-stress and *a* and *b* are the fitting parameters. *a =* −2.61 and *b* = 449.29 MPa for the red fitting line in Figure 13, corresponding to the specimens that were tested in air; *a* = −0.77 and *b* = 396.33 MPa for the black fitting line in Figure 13, corresponding to the specimens that were tested in Ar. In the range of the *σ*_0_ used, the black fitting line shows that *σ**_f_* decreased with the increase of *σ*_0_, which was also demonstrated by the length of microcracks under different *σ*_0_ values. It was also considered that the accumulated damage (microcracks) in each cycle increased with the increase of *σ*_0_. The fitting line corresponding to the tests in air lies above that corresponding to the tests in Ar, and the slope of the fitting line that approximates the tests in air is greater than the slope corresponding to the specimens that were tested in Ar. This indicates that in the range of the testing conditions, the healing effect that was induced by oxidation may have helped to improve thermal shock resistance, but such an improvement would have been reduced with the increase of the applied pre-stress. This effect can be attributed to the larger damage that was developed in each cycle of thermal shock under a larger *σ*_0_. The two fitting lines intersect at *σ*_0_
≈ 30 MPa, indicating that the *σ**_f_* of the specimens that were tested in air became inferior to that tested in Ar as *σ*_0_ ≥ 30 MPa.

The variations of *σ**_f_* against *σ*_0_ that was subjected to different thermal shock cycles are shown in Figure 14, including the result of the present work and our previous work [44]. This shows that the *σ**_f_* of all the tested specimens was lower than the *σ**_f_* of the specimen that did not undergo a thermal shock test. This is a different result from the previous work by Meng et al. [38,39], which might be ascribed to the different constituents of the used composite and the higher target temperature (2000 °C) in present study. The *σ**_f_* of the specimen that was subjected to repeated thermal shocks without pre-stress did not show a distinct tendency, as it was mainly affected by oxidation time or the thickness of oxide layer, and it also did not show distinct differences between the present and previous tests. However, once *σ*_0_ is applied, *σ**_f_* should, in general, decrease with the increase of *σ*_0_, i.e., the larger the *σ*_0_, the lower the *σ**_f_*.

## 4. Conclusions

We have presented a comprehensive investigation of the effects of pre-stress (*σ*_0_) and testing environment (in air and Ar atmospheres) on the residual flexural strength (*σ**_f_*) of a ZrB_2_–SiC–graphite composite that was subjected to repeated thermal shock.

The fractographs and backscattered electron images show that there were more pits in the specimens that were tested in Ar than that in the specimens that were tested in air, which could be ascribed to the protective glassy layer on the surfaces of the specimens that were tested in air. The average length of microcracks in the specimens that were tested in air were longer than those that were tested in Ar, which could be ascribed to the coupled effect of oxidation and thermal shock.

The fitting lines of the variations of *σ**_f_* against *σ*_0_ of the specimens that were tested in air and those that were tested in Ar had similar descending trends, but the former was above the latter and the descent slope of the former was larger than that of the latter. This can be attributed to the healing effect of the oxidation layer, but such effect would have been degraded with the increase of *σ*_0_, when damage accumulated in each cycle that was caused by coupled oxidation and thermal shock became larger than that caused by pure thermal shock.

The variations of *σ**_f_* against thermal shock cycles did not show a distinct tendency in the case of *σ*_0_ = 0 MPa, but they did show decent trends once *σ*_0_ was applied. In general, *σ**_f_* decreased with the increase of *σ*_0_, i.e., the larger the *σ*_0,_ the lower the *σ**_f_*. The *σ**_f_* of all the specimens that were subjected to repeated thermal shock testing was smaller than that of the specimens that were made of the as fabricated composite (489.4 MPa).

## Figures and Tables

**Figure 1 materials-13-00370-f001:**
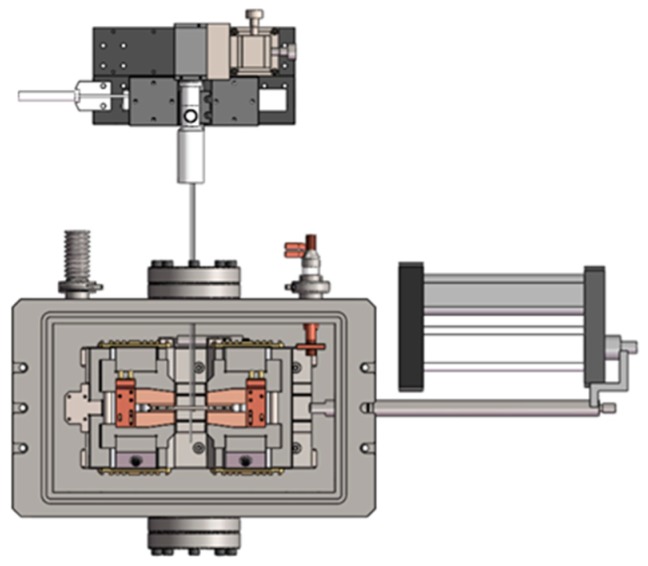
Schematic of the coupled thermal–mechanical material testing system.

**Figure 2 materials-13-00370-f002:**
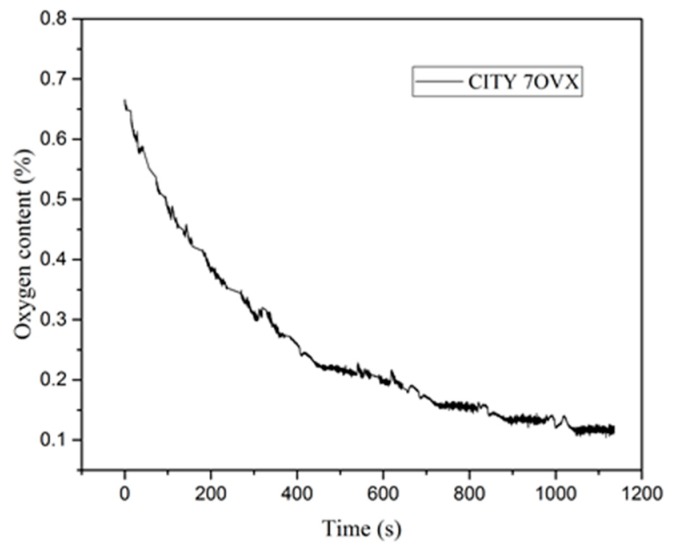
Oxygen content in sealed chamber during testing in Ar.

**Figure 3 materials-13-00370-f003:**
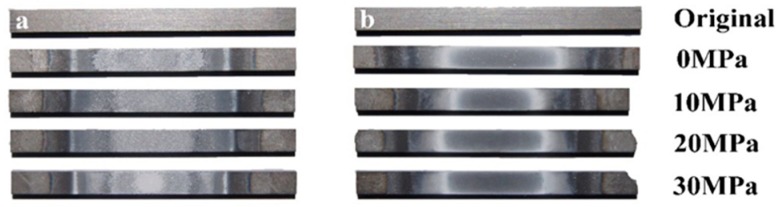
Surface macrographs of specimens that were subjected to thermal shock 60 times under different *σ*_0_ values (**a**) in air and (**b**) Ar.

**Figure 4 materials-13-00370-f004:**
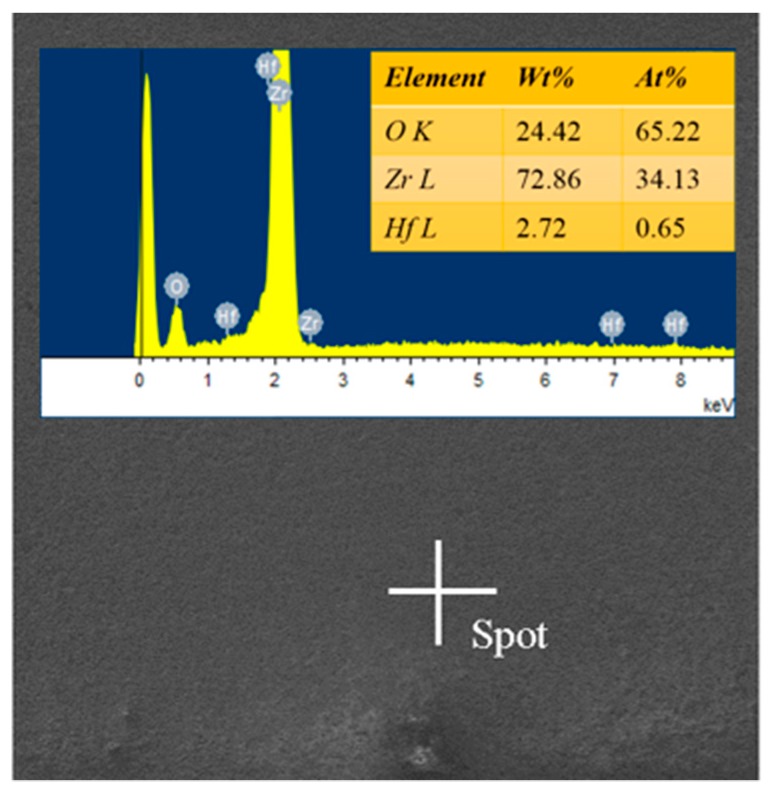
Energy dispersive spectroscopy (EDS) result of the constituents on the surface of the specimens that were tested in the Ar atmosphere.

**Figure 5 materials-13-00370-f005:**
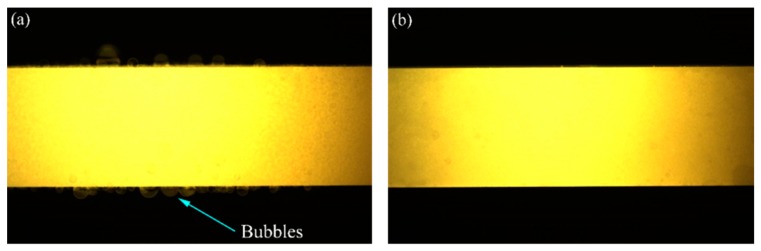
Surface images of specimens that were subjected to *σ*_0_ = 20 MPa at 2000 °C (**a**) in air and (**b**) Ar.

**Figure 6 materials-13-00370-f006:**
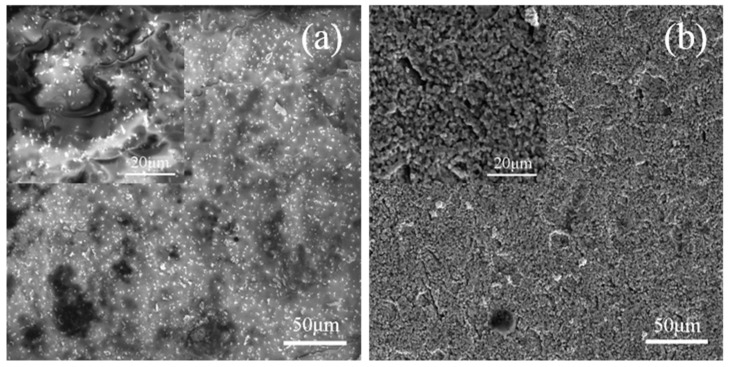
SEM photographs of specimens that were subjected to 60 cycles of thermal shock under *σ*_0_ = 20 MPa (**a**) in air and (**b**) Ar.

**Figure 7 materials-13-00370-f007:**
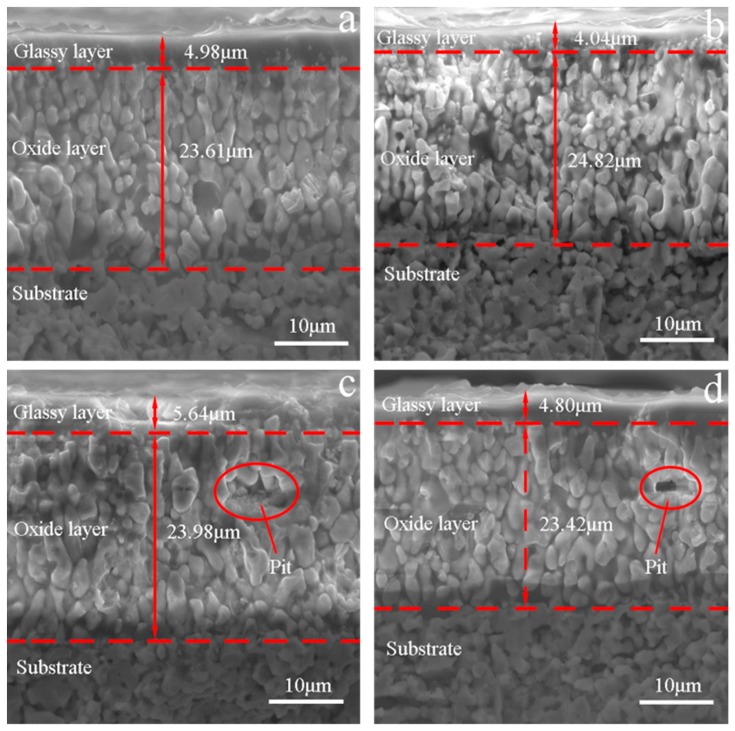
Fractured surface of the specimen that was subjected to 60 cycles of thermal shock in air (**a**) *σ*_0_ = 0 MPa, (**b**) *σ*_0_ = 10 MPa, (**c**) *σ*_0_ = 20 MPa, and (**d**) *σ*_0_ = 30 MPa.

**Figure 8 materials-13-00370-f008:**
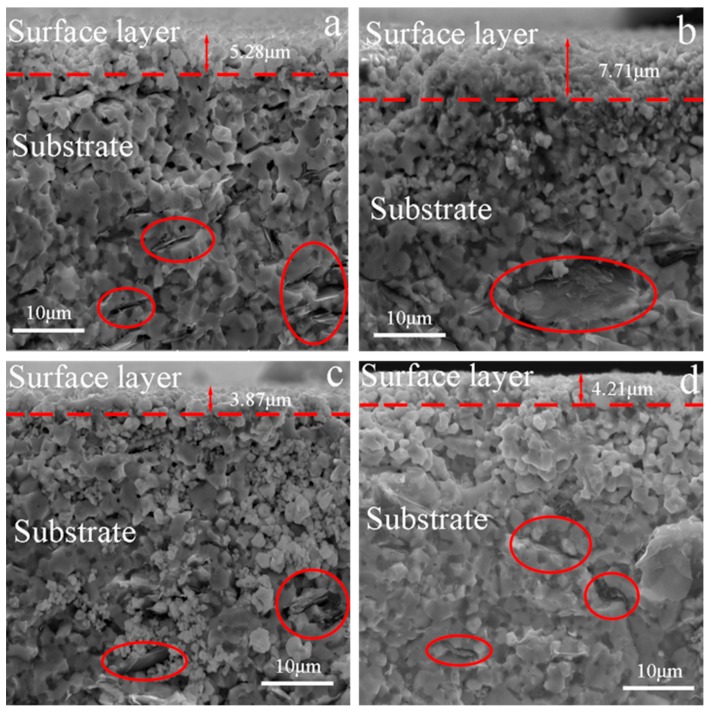
Fractured surface of the specimen that was subjected to 60 cycles of thermal shock in Ar (**a**) *σ*_0_ = 0 MPa, (**b**) *σ*_0_ = 10 MPa, (**c**) *σ*_0_ = 20 MPa, and (**d**) *σ*_0_ = 30 MPa.

**Figure 9 materials-13-00370-f009:**
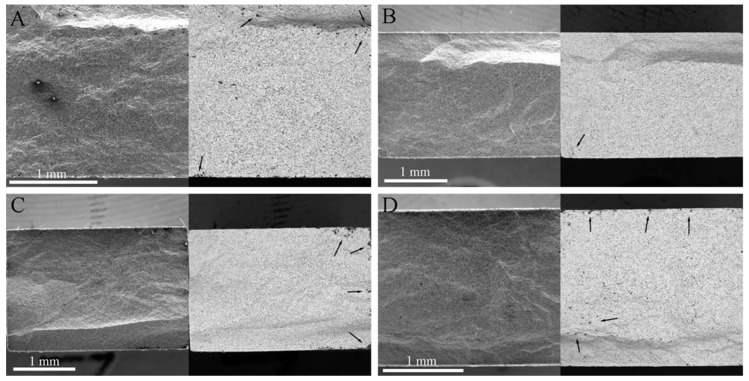
Fractographs and corresponding backscattered electron images of the ZrB_2_–SiC–graphite specimen tested. (**A**) ZSG0, (**B**) ZSG10, (**C**) ZSG20, and (**D**) ZSG30.

**Figure 10 materials-13-00370-f010:**
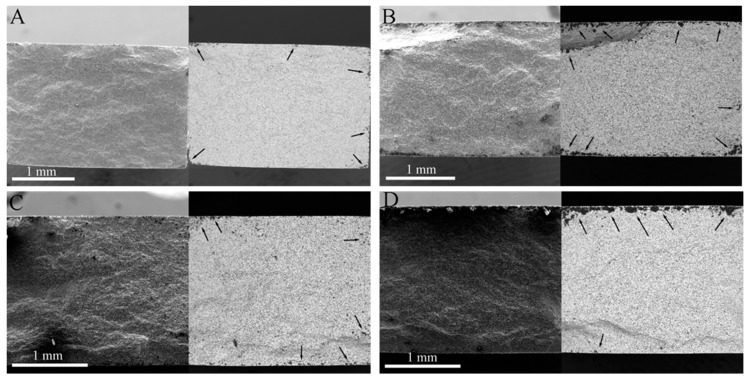
Fractographs and corresponding backscattered electron images of the ZrB_2_–SiC–graphite specimens that were tested. (**A**) ZSG0–Ar, (**B**) ZSG10–Ar, (**C**) ZSG20–Ar, and (**D**) ZSG30–Ar.

**Figure 11 materials-13-00370-f011:**
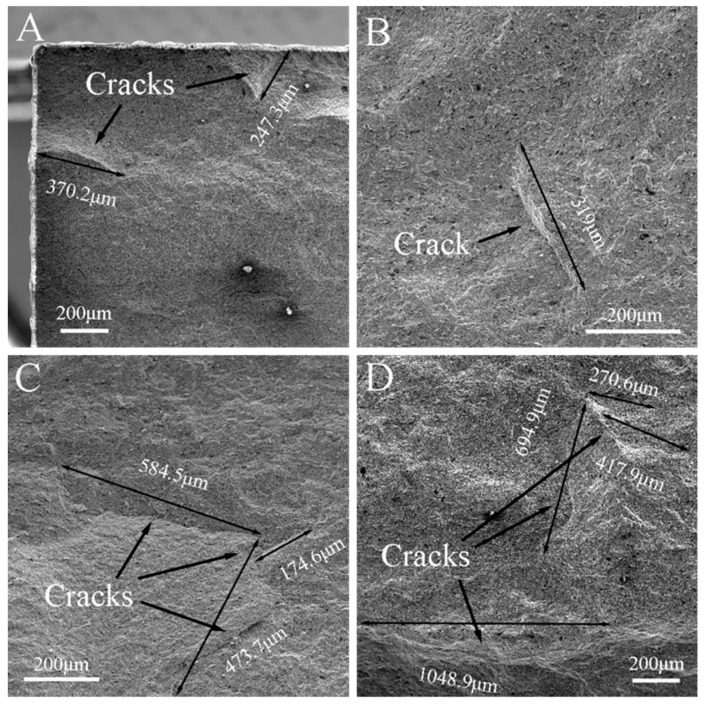
Microcracks in the fracture surface of the ZrB_2_–SiC–graphite specimens that were tested in air. (**A**) ZSG0, (**B**) ZSG10, (**C**) ZSG20, and (**D**) ZSG30.

**Figure 12 materials-13-00370-f012:**
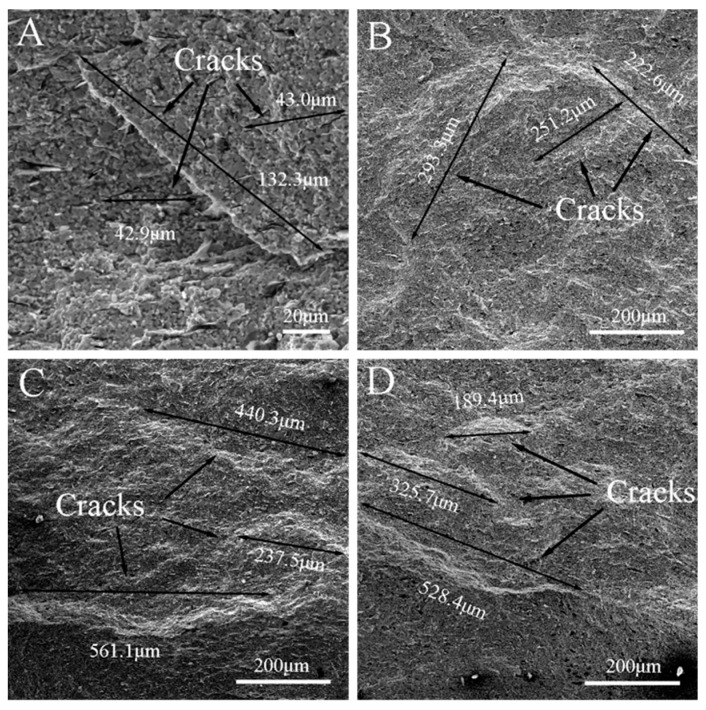
Microcracks in the fracture surface of the ZrB_2_–SiC–graphite specimen that was tested in Ar. (**A**) ZSG0–Ar, (**B**) ZSG10–Ar, (**C**) ZSG20–Ar, and (**D**) ZSG30–Ar.

**Figure 13 materials-13-00370-f013:**
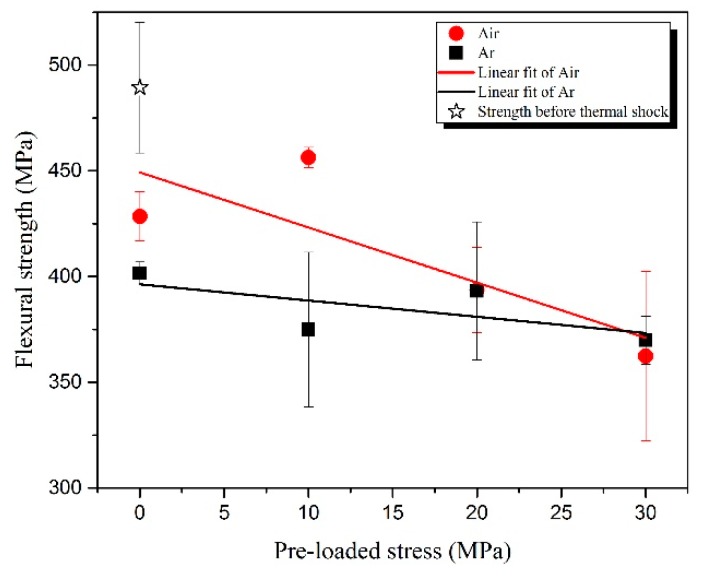
Residual flexural strengths of the specimens that were subjected to 60 cycles of thermal shock under different pre-stresses that were tested in the air and Ar atmospheres.

**Figure 14 materials-13-00370-f014:**
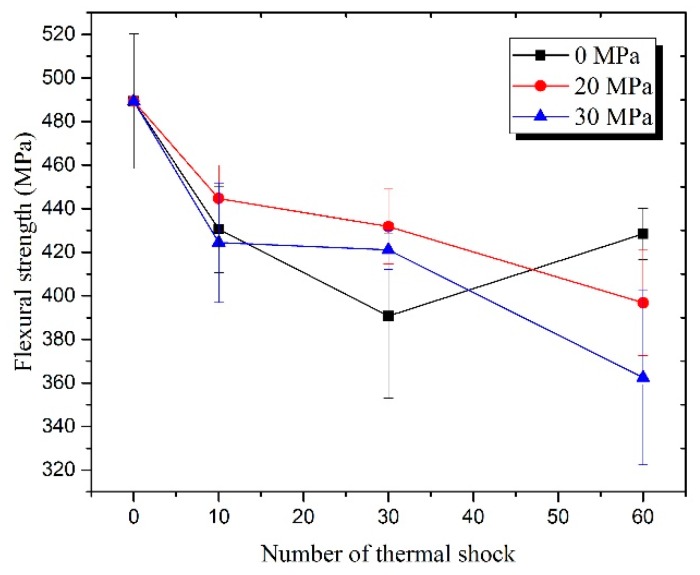
Residual flexural strengths of the specimens that were subjected to 0, 10, 30 and 60 cycles of thermal shock in air under different pre-stresses.

**Table 1 materials-13-00370-t001:** Conditions for repeated thermal shock tests.

Specimen and Test Name	Pre-Loaded Stress *σ*_0_ (MPa)	Gaseous Environment	Cycles
ZSG0	0	Air	60
ZSG10	10		
ZSG20	20		
ZSG30	30		
ZSG0-Ar	0	Ar	
ZSG10-Ar	10		
ZSG20-Ar	20		
ZSG30-Ar	30		

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
