# Peer review of "Repeated Thermal Shock Behavior of ZrB2–SiC–Graphite Composite under Pre-Stress in Air and Ar Atmospheres"

_materials, 2020, doi:10.3390/ma13020370_

Round 1

Reviewer 1 Report

Ref.: materials-684327

 Yue et al. have reported on the repeated thermal shock behavior of ZrB2-SiC-graphite composites under pre-stress in air and Ar atmospheres. Although the results are convincing, the manuscript may not be accepted in the present form. It needs compulsory major revision in light of the following comments/suggestions.

Comments/suggestions:

The same authors of group have already published their results in [ref. 35 and 37] with same materials system. Only, the main difference from their previous studies is that in the present work TSR was carried in Ar and for 60 cycles. Previous works report for different cycles and for air atmosphere for the same materials system. In fact, figures 1 and 3 are copied and reproduced from their previous publications. Figure 4 looks almost similar with previous works. There is no need to provide the same figures. Just give the necessary references. It is very hard to understand the differences between microstructural and flexural strength results when compared with their previous works. For example, in the case of flexural strength data, the values are very different for the reported materials system in the case of zero MPa pre-stress in the present and previous works. I recommend writing and explaining their results in the light of their previous works. They need to mention flexural strength of as fabricated and how it varies with respect to changing the parameters like pre-stress, atmosphere and heating rates etc. There is a need to explain their results systematically. In the abstract, the authors write “the thermo-chemo-mechanical…”, whose meaning is difficult to   Whether is it chemical?

Reviewer 2 Report

The paper reports some interesting results on the effect of different tensile stresses and different environments on the thermal shock resistance of ZrB2-20vol.%SiC-15vol%graphite composites. The paper is well written, the experiments are well planned and performed. There are just some minor issues, which should be addressed before publication:

The novelty of the work is not sufficiently described in the introduction part. In the introduction, the authors mentioned that the thermal shock resistance of the same material composition has already been investigated and reported. Moreover, the effect of different stresses on the thermal shock resistance of the same materials was investigated by the same group of authors (reference 38). What is the novelty of the present paper, compared to the earlier works? The authors are advised to emphasize that at the end of introduction... The information on the homogenization technique used to prepare powder mixture and the information on the loading rate during the flexural testing will benefit the readers... Page 5, lines 163-164: “The constituent of the surface of specimens tested in air is mainly SiO2.....” At the end of the same paragraph, the authors said the surface layer consists of several oxidation products, such as borosilicate glass and ZrO2 particles.... That should be consistent to avoid feeling of contradiction. Page 7, lines 207-208: “These glassy layers are mainly composed of solidified SiO2 and B2O3, ....”. Later in the same paragraph, the authors said B2O3 will evaporate at the temperatures above 1100 C. If it evaporates at these temperatures, it should not be present in the glassy layer of the material tested at 2000 C, should be? In addition, the evaporation of gaseous B2O3 should leave behind a porous glassy layer, however the layer shown in Figure 8 is dense and continuous. The authors are advised to improve this paragraph and make it more clear to avoid misunderstanding. The main aim of the paper is to report the resistance of ZrB2-SiC-graphite materials against multiple thermal shocks. However, the effect of graphite addition on the thermal shock in different environments is not discussed whatsoever, thereby it is not clear what contribution the graphite addition into the ZrB2-SiC material system actually is...

Reviewer 3 Report

The paper "Repeated thermal shock behavior of ZrB2-SiC-graphite composite under prestress in air and Ar atmospheres" falls within the scope of Materials Journal and shows technical relevance.

In this paper, the authors investigate experimentally the effects of heating atmosphere and prestress on the mechanical property of ZrB2-20 vol% SiC-15 vol% graphite (ZSG) composite.

The material is publishable but requires improvement. In this sense, there are some suggestions on the attached paper that should be addressed before publishing.

Suggestion 01

In the view of this reviewer, in the last paragraph of the introduction section, the authors fail to clarify the main objectives and limitations and to point out the major contributions of the study.

Suggestion 02

In the same way, in sub-section 2.1, the self-developed coupled thermal-mechanical material testing system should be shown in a schematic way, clarifying the authors’ original inputs and the novelty degree of the performed methodology.

Suggestion 03

Given the huge research field involved, there is needed a literature review improvement. In this regard, the following papers are recommended for consideration:

Mehdi Shahedi Asl et Al. (2018), in Ceramics International; Shanbao Zhou et Al. (2010), in Materials Chemistry and Physics; Shuqi Guo (2013), in Ceramics International; Elisa Padovano et Al (2018), in Ceramics International; X.H. Zhang et Al (2009), in Scripta Materialia and Shubin Wang et Al (2013), in Journal of Refractory Metals and Hard Materials

Suggestion 04

The conclusions section should be rewritten as data must be related to the main research questions, (which are not clearly identified in the current version of the paper), in order to validate the results themselves as well as the proposed methodology.

Reviewer 4 Report

Dear authors,

The manuscript provides important contributions to the research concerning the effects of the thermal shock from room temperature until to 2000 °C, on the residual flexural strength.

Your manuscript work is very interesting from experimental point of view and the conclusions are adequately supported by the data presented. But, the paper would become much more interesting if the authors add the results regarding the effects of the thermal cycles on the modulus of elasticity in bending. The flexural properties must be also compared with the flexural properties corresponding to the reference specimens (specimens that are not subjected to thermal cycles before bending).

I would like to recommend the publication of this paper after the authors will make some improvements and introduce additional data like it is recommended in the following.

Section “Experimental procedure”:

-Lines 120-121: Authors should give more details about the equipment used to manufacture the composite by hot-pressing.

-Has a mixer been used to homogenize the mixture of the three powders before manufacturing by hot pressing? Please, add details concerning the homogenization.

-Lines 123-124: The details about the flexural test should be moved in a new sub-section “2.4. Flexural test” inserted after the section “2.3. Repeated thermal shock tests”. In this sub-section, the authors must give more details about the type of the equipment used in bending test, the standard used in bending test of the composite involved, speed of loading.

-What is the reason for which the authors considered 60 cycles in thermal shock tests? Is the thermal shock test in accordance with any standard recommended for this kind of composites involved in this research?

Section “3. Results and discussion”:

-The section could be divided into two sub-sections: “3.1. Surface analysis” and “3.2. Residual flexural properties”.

-Authors should consider also the effects of the thermal shock on the modulus of elasticity in bending. The paper will be much more interesting after the authors will include in this paper the graph of the variation of the modulus of elasticity after 60 thermal cycles.

-All results obtained in bending on the specimens subjected to thermal cycles must be compared with the results obtained in bending tests on the reference specimens (specimens that are not subjected to thermal cycles before bending). Please, add to Figure 14 the results obtained on reference specimens.

-Authors must give the equations of the two lines shown in Figure 14, which approximate the experimental data for the two sets of specimens tested in air and Ar atmosphere, respectively.

-Line 341: I recommend more care in expression. For example, the expression “the former is steeper than the latter” should be replaced with “The slope of the line that approximates the tests in air, is greater than the slope corresponding to the specimens tested in Ar”.

Round 2

Reviewer 1 Report

The revised manuscript may be accepted.

Reviewer 3 Report

In general, recommendations have been considered in the paper, so the current version is considered acceptable for publication.